# Iron Deficiency and Sleep/Wake Behaviors: A Scoping Review of Clinical Practice Guidelines—How to Overcome the Current Conundrum?

**DOI:** 10.3390/nu16152559

**Published:** 2024-08-03

**Authors:** Scout McWilliams, Olivia Hill, Osman S. Ipsiroglu, Stefan Clemens, Alexander Mark Weber, Michael Chen, James Connor, Barbara T. Felt, Mauro Manconi, Andre Mattman, Rosalia Silvestri, Narong Simakajornboon, Susan M. Smith, Sylvia Stockler

**Affiliations:** 1H-Behaviours Research Lab (Previously Sleep/Wake-Behaviours Research Lab), BC Children’s Hospital Research Institute, Department of Pediatrics, University of British Columbia, Vancouver, BC V5Z 4H4, Canada; scout.mcwilliams@bcchr.ca (S.M.); olivia.hill@bcchr.ca (O.H.); sstockler@cw.bc.ca (S.S.); 2Divisions of Developmental Pediatrics, Child and Adolescent Psychiatry and Respirology, BC Children’s Hospital, Department of Pediatrics, University of British Columbia, Vancouver, BC V6T 1Z4, Canada; 3Department of Physiology, Brody School of Medicine, East Carolina University, Greenville, NC 27834, USA; clemenss@ecu.edu; 4Department of Pediatrics, University of British Columbia, Vancouver, BC V6T 1Z4, Canada; aweber@bcchr.ca; 5BC Children’s Hospital Research Institute, Vancouver, BC V5Z 4H4, Canada; 6Department of Pathology and Laboratory Medicine, University of British Columbia, Vancouver, BC V6T 1Z4, Canada; michael.chen@islandhealth.ca (M.C.); amattman@providencehealth.bc.ca (A.M.); 7Department of Neurosurgery, Penn State Hershey Medical Center, Hershey, PA 17033, USA; jconnor@pennstatehealth.psu.edu; 8Department of Pediatrics, University of Michigan, Ann Arbor, MI 48109, USA; truefelt@med.umich.edu; 9Sleep Medicine Unit, Neurocenter of the Southern Switzerland, Regional Hospital of Lugano, Faculty of Biomedical Sciences, Università della Svizzera Italiana, 6900 Lugano, Switzerland; mauro.manconi@eoc.ch; 10Department of Neurology, University of Bern, 3012 Bern, Switzerland; 11Department of Clinical and Experimental Medicine, Sleep Medicine Center, University of Messina, Azienda Ospedaliera Universitaria “Gaetano Martino”, 98122 Messina, Italy; rosalia.silvestri@unime.it; 12Sleep Center, Cincinnati Children’s Hospital Medical Center, Cincinnati, OH 45229, USA; narong.simakajornboon@cchmc.org; 13Department of Nutrition, UNC-Nutrition Research Institute, University of North Carolina at Chapel Hill, Kannapolis, NC 28081, USA; susan_smith@unc.edu; 14Division of Biochemical Diseases, Department of Pediatrics, BC Children’s Hospital, University of British Columbia, Vancouver, BC V6T 1Z4, Canada

**Keywords:** attention deficit hyperactivity disorder, brain iron, central iron deficiency, clinical practice guidelines, iron deficiency, restlessness, restless legs syndrome, sleep disorders

## Abstract

Current evidence suggests that iron deficiency (ID) plays a key role in the pathogenesis of conditions presenting with restlessness such as attention deficit hyperactivity disorder (ADHD) and restless legs syndrome (RLS). In clinical practice, ID and iron supplementation are not routinely considered in the diagnostic work-up and/or as a treatment option in such conditions. Therefore, we conducted a scoping literature review of ID guidelines. Of the 58 guidelines included, only 9 included RLS, and 3 included ADHD. Ferritin was the most frequently cited biomarker, though cutoff values varied between guidelines and depending on additional factors such as age, sex, and comorbidities. Recommendations surrounding measurable iron biomarkers and cutoff values varied between guidelines; moreover, despite capturing the role of inflammation as a concept, most guidelines often did not include recommendations for how to assess this. This lack of harmonization on the interpretation of iron and inflammation biomarkers raises questions about the applicability of current guidelines in clinical practice. Further, the majority of ID guidelines in this review did not include the ID-associated disorders, ADHD and RLS. As ID can be associated with altered movement patterns, a novel consensus is needed for investigating and interpreting iron status in the context of different clinical phenotypes.

## 1. Introduction

Iron deficiency (ID) is the most common micronutrient deficiency in the world and one of the biggest contributors to the global prevalence of anemia. Children, females of reproductive age, and those living in low-income countries or of low socioeconomic status are disproportionately affected by ID [1]. Besides its essential roles in hemoglobin formation and hematopoiesis, energy generation, inflammation, remission, and healing, iron has numerous functions within the central nervous system, for example, as a cofactor for various enzymes, particularly within the dopaminergic and glutamatergic systems. The effects of central ID (CID) on measurable neurotransmitters, receptor levels, excitability, and behaviors have been illustrated in Figure 1 [2,3]. Further, animal studies have proven the importance of iron in brain development and functioning [4]. In humans, maternal ID in pregnancy, particularly in the first trimester, has been associated with low birth weight and preterm delivery, as well as an increased risk of intellectual disability, autism spectrum disorder (ASD), and attention deficit hyperactivity disorder (ADHD) in the offspring [5,6].

Importantly, ID can occur in the absence of anemia and is referred to as non-anemic iron deficiency (NAID). While much of the focus has historically been on ID anemia (IDA), recent evidence has shown that NAID can have significant consequences on health and wellbeing. NAID has been associated with neurodevelopmental disorders such as ADHD and a number of sleep disorders, such as restless legs syndrome (RLS), periodic limb movements in sleep (PLMS), and restless sleep disorder (RSD), all of which may profit from iron supplementation as a therapeutic intervention [7,8,9,10,11]. These disorders typically present with restlessness in sleep and wake and during transitions from wake to sleep states. While the specific mechanisms by which NAID contributes to symptoms in the aforementioned conditions are not fully understood, current evidence in humans by imaging studies suggests that CID may be an important factor (Table 1), which may not be reflected by peripheral iron levels.

Rodent models utilizing diet modifications show the causal links between ID and alterations in dopamine transmission that are consistent with a hyperdopaminergic hypothesis of RLS [2]. Despite this strong evidence of possible biological interactions, there is a general paucity of ADHD animal models that incorporate ID or findings from RLS models. Intriguingly, data from the spontaneously hypertensive rat model present the preclinical characteristics of both RLS and ADHD [12]. As hypertension is regulated in part by the autonomic nervous system and iron levels [13], this may indicate an additional layer of complexity in the regulation of ADHD and RLS and further justifies the terminology of CID syndromes to facilitate more in-depth clinical phenotyping of restlessness, as well as blood work investigations for the treatment of sleep disorders such as RLS, PLMS, and RSD and also ADHD [7,10,14]. The association between iron homeostasis and restlessness in sleep and wake states raises questions about which iron biomarkers should be measured, the timing of such measurements, and how the results should be interpreted.
nutrients-16-02559-t001_Table 1Table 1Summary of MRI brain iron studies in ADHD and RLS. ADHD: attention deficit hyperactivity disorder; MFC: magnetic field correlation; mo: months; PLMS: periodic limb movements of sleep; QSM: quantitative susceptibility mapping; RLS: restless legs syndrome; ROI: regions of interest; SF: serum ferritin, Y: years.AuthorStudy PopulationMethodResultsConditionParticipant Number Age (Y):RangeMeanAdisetiyo et al., 2014 [15]ADHD*n* = 22 ADHD*n* = 27 controls8–1812.713.3MRI imaging relaxation rates (R2, R2*, R2′) and magnetic field correlation (MFC) in the globus pallidus, putamen, caudate nucleus, and thalamus-R2, R2*, R2′-MFC-No difference in R values-Lower MFC in ADHD (lower brain iron)Adisetiyo et al., 2019 [16]ADHD*n* = 30ADHD*n* = 29 controls 8–1814.013.9-R2*-MFC-No difference in R2* or MFC-Increased R2* and MFC (increased brain iron) with psychostimulant use duration in ADHD more than with ageAllen et al., 2001 [17]RLS*n* = 5 RLS*n* = 5 controls 66.266.4-R2′-Lower R2′(lower brain iron) in RLS and in proportion to RLS severityAstrakas et al., 2008 [18]RLS*n* = 25 RLS*n* = 12 controls 55–8266.554–8965.7-T2-Higher T2 (lower brain iron) in RLSBeliveau et al., 2022 [19]RLS*n* = 72 RLS*n* = 72 controls46–5951.9(median)51.0 (median)-R2, R2′, and R2*-QSM-Higher R and QSM values (increased iron brain iron) in RLS Cortese et al., 2012 [20]ADHD*n* = 18ADHD*n* = 9controls*n* = 9 psychiatric controls 118.8 mo120.8 mo123.5 mo-T2*-Higher T2* (lower brain iron) in ADHD-SF and T2* values did not correlate significantly in most regionsEarley et al., 2006 [21]RLS*n* = 22 early-onset RLS*n* = 19 late-onset RLS*n* = 39 controls57.167.460.5-R2′-Lower R2′ (lower brain iron) in early-onset RLS symptoms, but not late-onset RLSGodau et al., 2008 [22]RLS*n* = 6 RLS*n* = 19 controls 47–686059-T2-Higher T2 (lower brain iron) in RLSHasaneen et al., 2017[23]ADHD*n* = 17 ADHD*n* = 18 controls 6–158.48.5-R2*-Lower R2* (lower brain iron) in ADHD which correlated with ADHD type but not with ADHD severityKnake et al., 2010 [24]RLS*n* = 12 RLS*n* = 12 controls 43–4658.541–7456.8-T2-No difference in T2 valuesLi et al., 2016 [25]RLS*n* = 39 RLS*n* = 29 controls 58.457.9-QSM-Lower magnetic susceptibility (lower brain iron) in RLS and possible connection to PLMSMargariti et al., 2012 [26]RLS*n* = 11 RLS*n* = 11 controls 48–7055.342–7356.1-T2-Lower T2 (higher brain iron) in RLSMoon et al., 2014 [27]RLS*n* = 37 RLS*n* = 20 early-onset RLS*n* = 17 late-onset RLS*n* = 40 RLS controls*n* = 20 early-onset controls*n* = 20 late-onset controls50.358.147.059.4-T2-Higher T2 (lower brain iron) in late-onset RLS, but not early-onset RLSMoon et al., 2015 [28]RLS*n* = 37 RLS*n* = 40 controls 53.853.2-R2, R2*, and R2′-Relaxometry and ROI determination methods significantly influenced the outcome of brain iron estimatesRizzo et al., 2013 [29]RLS*n* = 15 RLS*n* = 15 controls 51.051.0-Phase from gradient-echo scan-Higher phase values (lower brain iron) in RLS


Historically, ID was diagnosed solely based on hematologic markers such as hemoglobin in conjunction with, for example, mean corpuscular volume and mean corpuscular hemoglobin. Over the last two decades, however, iron-specific markers, most notably serum ferritin (SF), have become the mainstay for diagnosing NAID and IDA. Since 2001, the World Health Organization (WHO) has recommended SF as the primary biomarker for the assessment of iron status [30]. SF is an iron storage protein and acute phase reactant that is present in all cells. While low SF concentrations indicate low iron stores, normal or elevated SF does not exclude ID in cases of infection or inflammation [1].

We carried out a scoping literature review with the aim to investigate clinical ID guidelines, specifically looking at recommendations for diagnosis, such as iron biomarkers, their cutoff values, and the role of inflammation when interpreting laboratory results. Our secondary aim was to identify whether common disorders presenting with restlessness such as ADHD and RLS have been incorporated into the clinical guidelines, as diagnosis and treatment can have significant implications for the affected individuals. A scoping review was the agreed-upon methodology given the exploratory nature of this project and to facilitate the identification of gaps in clinical knowledge.

## 2. Materials and Methods

This scoping review was carried out in accordance with the PRISMA-Scr Checklist (Appendix A, Table A1). A protocol for this review does not exist. Two reviewers (SM and OH) were involved in identifying guidelines for inclusion. The initial search was carried out on 27 June 2020 in CINAHL, Embase, and Medline with no date restrictions and was updated on 7 April 2023. The search strategy included variations of the terms “iron deficiency” and “guideline”; the detailed search strategy is laid out in Table 2. Covidence, a web-based collaboration software platform that streamlines the production of systematic and other literature reviews (https://www.covidence.org/, accessed on 18 May 2024), was employed for the selection and de-duplication process. The search was updated in 2023. Additional guidelines were identified by searching Trip Medical Database (https://www.tripdatabase.com/, accessed on 18 May 2024) as well as conducting Google searches and checking websites of medical organizations. Two reviewers (SM and OH) carried out data extraction and organized data into a spreadsheet. A third reviewer (OI) was available to oversee this process.

### 2.1. Inclusion Criteria

Guidelines were included if (1) they were general ID guidelines, defined as those which targeted a general population (e.g., adults, elderly, children, ethnic groups), pregnancy-specific ID guidelines, and disease-specific guidelines (e.g., chronic kidney disease (CKD)); (2) the guideline or consensus paper was created by/on behalf of a larger governing body (e.g., international, national, or regional organizations/societies); (3) they were available in English-language.

### 2.2. Exclusion Criteria

Guidelines were excluded if they were (1) opinion papers or guidelines published by authors not affiliated with a larger governing body; (2) reviews of clinical guidelines.

### 2.3. Data Analysis

Three reviewers (SM, OH, and OI) were involved in analyzing the extracted data. A qualitative analysis was carried out by reviewing the following categories:1Population defined by age, pregnancy status, and medical conditions. Guidelines were organized into three categories: (1) general ID, (2) ID in pregnancy, and (3) disease-specific ID.General ID guidelines were defined as those guidelines which could be applied to a general population and which may have included specific subpopulations within the guideline.Disease-specific ID guidelines were defined as those guidelines which dealt with only a specific population, namely chronic disease populations, in which the diagnosis and management of ID is different from general ID guidelines.1.1Year and country of publication.2Associated clinical presentations, conditions, diagnoses, and risk factors for ID.2.1If ADHD and/or RLS were included as either signs/symptoms or as being associated with ID. Guidelines that used broad terminology such as “behavioral disturbances” or “sleep disturbances” without specifying the aforementioned conditions were not classified as having included ADHD and/or RLS.3.Suggested cutoff values for SF, taking into account age- and sex-specific cutoff values.3.1Additional iron and hematologic biomarkers included in the guidelines. Examples of iron biomarkers (other than SF) are serum iron and transferrin saturation, while hematologic biomarkers include hemoglobin and mean corpuscular volume/mean corpuscular hemoglobin.

## 3. Results

A total of 58 ID guidelines were identified; 20 identified through the electronic databases, 36 identified through searching websites of international, national, and regional health organizations, and 2 identified on Trip Medical Database. A total of 33 were general ID guidelines, 10 were pregnancy-specific, and 15 were disease-specific (Figure 2; Table A2, Table A3 and Table A4). The included guidelines were published between 1989 and 2023 with the majority published in the 2010s (*n* = 36) [31,32,33,34,35,36,37,38,39,40,41,42,43,44,45,46,47,48,49,50,51,52,53,54,55,56,57,58,59,60,61,62,63,64,65,66]. Thirteen were published in the 2020s [67,68,69,70,71,72,73,74,75,76,77,78,79], six in the 2000s [30,80,81,82,83,84], one in the 1990s [85], and one in the 1980s [86]. One guideline had no publication date [87].

### 3.1. Guideline Development Methodology

A total of 34/58 guidelines did not provide any explanation of the methods employed for guideline development. Of the guidelines that did describe the methods, 20 provided a grading of the evidence. To assess the certainty of evidence and strength of recommendations, the most employed method was the GRADE approach, used in eight guidelines [50,53,60,62,68,71,72,74]. Other methods included the Oxford Centre for Evidence Based Medicine (*n* = 2) [63,76], Scottish Intercollegiate Guidelines Network (*n* = 1) [33], National Health and Medical Research Council (*n* = 1) [38], National Comprehensive Cancer Network (*n* = 1) [66], Infectious Disease Society of America—US public health service grading system (*n* = 1) [49], Canadian Task Force on Preventive Health Care to grade recommendations (*n* = 1) [44], British Committee for Standards in Haematology (*n* = 1) [45], and Minds2014 (*n* = 1) [59]. Three guidelines did not specify which grading system was used [80,83,84].

### 3.2. Types of Iron Deficiency

A total of 24/33 general and 4/10 pregnancy-specific guidelines focused only on IDA [30,32,33,34,36,37,38,39,40,42,43,46,52,53,54,55,57,58,70,71,73,76,77,80,81,85,86,87], while 9 general and 6 pregnancy-specific guidelines included information on both IDA and non-anemic iron deficiency (NAID) [31,35,41,44,45,48,51,64,65,67,68,72,74,75,78].

### 3.3. Biomarkers

#### 3.3.1. Serum Ferritin and Other Iron Biomarkers

A total of 54/58 guidelines recommended measuring SF and provided cutoff values [30,31,33,34,35,37,38,39,40,41,42,43,44,45,46,47,48,49,50,51,52,53,54,55,56,57,58,59,60,61,63,64,65,66,67,68,69,70,71,72,73,74,75,76,77,78,79,80,81,82,83,84,85,87]. Three guidelines did not include SF [32,36,86], and one included SF but did not include cutoff values [62]. Besides SF, the most commonly suggested iron-specific markers were transferrin saturation (*n* = 45), serum iron (*n* = 29), total iron binding capacity (*n* = 27), and soluble transferrin receptor (*n* = 21) (Figure 3 and Figure 4). Figure 5 and Figure 6 outline the SF cutoff values recommended by the included guidelines.

##### General Iron Deficiency Guidelines

SF cutoff values ranged from 10 ug/L to 100 ug/L in general ID guidelines. Two general ID guidelines had differing SF cutoff values for females and males, whereby females had lower SF cutoff values than males (<13 ug/L females; <30 ug/L males) [42,54]. Eighteen general ID guidelines had child-specific SF cutoff values shown in Figure 5 [30,31,35,37,38,40,41,42,43,48,52,58,64,65,72,73,74,76].

##### Iron Deficiency in Pregnancy

SF cutoff values ranged from <10 ug/L to <30 ug/L. Two guidelines had SF cutoff values of <15 ug/L but noted that treatment should be initiated when SF <30 ug/L [53,77]. One guideline recommended that, in non-anemic pregnant females with more than 16 weeks gestation, iron supplementation be commenced when SF <60 ug/L [78].

##### Iron Deficiency in Disease-Specific States

The most common condition was CKD (*n* = 5) [50,59,82,83,84] followed by cancer (*n* = 4) [49,62,66,69], inflammatory bowel disease (IBD) (*n* = 1) [63], intestinal rehabilitation (*n* = 1) [79], and heart failure (*n* = 1) [56]. One guideline included three distinct diseases (congestive heart failure (CHF), CKD, and IBD) [61]. One guideline was designed for use in post-operative patients [47]. In addition, one guideline was specifically looking at functional iron deficiency (FID), with an emphasis on patients with CKD [60].

SF cutoff values ranged from <12 ug/L to <800 ug/L. Higher cutoff values for specific subpopulations were suggested in eight disease-specific guidelines; for individuals with inflammation/anemia of chronic disease (*n* = 2) [63,69], on hemodialysis (*n* = 2) [60,84], those receiving an erythropoiesis stimulating agent (ESA) (*n* = 3) [50,59,83], and those on hemodialysis and receiving an ESA (*n* = 1) [82]. Two guidelines included FID [60,66], for which the following SF cutoff values were suggested: <100 ug/L (non-hemodialysis patients) and <200 ug/L (hemodialysis patients) [60], <800 ug/L [66]. One disease-specific ID guideline did not provide a SF cutoff value, though SF measurement was still suggested [62].

#### 3.3.2. Hematologic Biomarkers

Hemoglobin measurement was recommended in 56 ID guidelines. Of these 56, two guidelines used hemoglobin [32,36], and one used hemoglobin and hematocrit [86] as the only recommended biomarker(s) for detecting ID/IDA. One guideline did not explicitly state that hemoglobin should be measured in the context of diagnosing ID in heart failure [56]. Similarly, the WHO guideline on the use of ferritin did not include hemoglobin, though this guideline was focused on SF rather than ID/IDA as a broader concept [74]. A detailed overview of all hematologic biomarkers is shown in Figure 3 and Figure 4.

#### 3.3.3. Biomarkers of Inflammation

A total of 25/58 ID guidelines recommended the concomitant measurement of CRP with SF [31,37,38,39,43,45,48,49,50,51,54,55,57,58,61,63,64,69,70,71,72,73,74,75,79]. Of these, two guidelines from the WHO recommended both CRP and α−1 acid glycoprotein (AGP) [43,74]. An additional 23 guidelines acknowledged the role of SF as an acute phase reactant that can increase in the presence of inflammation and/or infection, though measurement of CRP or another inflammatory marker was not explicitly stated [30,33,34,35,40,41,42,44,46,47,53,56,59,60,65,67,68,76,78,81,83,85,87].

### 3.4. Iron Deficiency, ADHD, and RLS

Three guidelines included ADHD (two general, one disease-specific) [35,42,63] and nine guidelines included RLS (four general; three pregnancy-specific, two disease-specific) [31,42,53,58,61,63,68,71,75] as signs/symptoms of ID or as associated conditions. Of these, one general and one disease-specific guideline included both ADHD and RLS [42,63]. Importantly, the BC Clinical Practice Guideline (Canada) included a SF cutoff value of ≤75 ug/L for iron supplementation in individuals with RLS [31].

## 4. Discussion

### 4.1. Iron Deficiency

ID is a paraphysiological state that precedes anemia and is associated with restlessness that can affect day- and night-time behaviors. Recent research suggests that ID, and more specifically CID, may be a precipitating factor in conditions that can present with overall restlessness, such as ADHD and RLS, which are two of the most frequently encountered neurobehavioral and neurologic conditions, with a prevalence of 7% and 4–14%, respectively [89,90]. We reviewed ID guidelines to determine the recommended iron biomarkers, their cutoff values, the role of inflammation when interpreting laboratory results, and whether these common disorders were incorporated.

In this scoping review, we identified 58 ID guidelines published across five decades and 11 countries, including 6 international guidelines. Included in this analysis were three guidelines from the WHO published in 2001, 2017, and 2020, respectively [30,43,74]. Our findings demonstrate that (1) SF is the primary biomarker for ID, though cutoff values vary between guidelines, (2) ADHD and RLS have seldom been incorporated as associated conditions or signs of ID, and (3) inflammation needs to be taken into consideration (e.g., by measuring CRP and completing a physical examination) when interpreting SF. The majority of guidelines in this review focused on IDA, while a smaller proportion included information on NAID with a preponderance to more recently published guidelines. This likely reflects the increasing knowledge that ID, even in the absence of anemia, can have adverse consequences on health [91].

### 4.2. Serum Ferritin

A total of 54/58 guidelines included in this review recommended SF as one of the primary biomarkers for diagnosing ID. However, cutoff values varied between guidelines and based on additional factors such as age or sex. Over the past two decades, the WHO guidelines have recommended SF as the primary marker of ID and utilized the same cutoff values during this time period [30,43,74]. However, there is clinical controversy suggesting that the cutoff values by the WHO may actually be too low, and proposals for higher cutoff values have been made. Based on data from the US National Health and Nutrition Examination Survey, utilizing information from 2569 children and 7498 non-pregnant women, the cross-sectional study by Mei et al. concluded that SF cutoff values of 20 ug/L for children and 25 ug/L for non-pregnant women would be more appropriate [92].

Due to the role of SF as an acute phase reactant, the utilization of additional biomarkers plays a key role in the identification of iron-depleted states. The two most recent WHO guidelines recommend screening for ID with SF and, when appropriate, the concomitant measurement of CRP and/or AGP [43,74]. While CRP is a well-recognized and widely available marker of inflammation, in our experience, AGP is less frequently used in clinical practice. In comparison to CRP, the rise in AGP is slower but remains elevated for a longer time period after the initial inflammatory event [93].

### 4.3. ID-Associated Conditions

In two recent scoping reviews investigating the role of ID in neurodevelopmental disorders and in sleep, respectively, 22 out of 30 studies demonstrated an association of ID and ADHD [7], and 24 out of 47 studies showed a positive association between ID and RLS [10]. The inconsistency in the results of the studies included in these scoping reviews may be explained by the heterogeneity of methodologies and missing the role of inflammation as a factor affecting iron homeostasis, indicating the need for a harmonization of the approach to ID in these populations. The association between ID and conditions presenting with restlessness is hypothesized to be due, in part, to the role of brain iron.

In the current scoping review, nine studies mentioned RLS and only three mentioned ADHD as being associated with ID. These results are unsurprising given that studies implicating ID, and more specifically CID, are relatively recent, but this highlights the knowledge gaps that exist in current clinical practice. While there is limited evidence to demonstrate the effect of iron supplementation on brain iron levels directly, iron supplementation has been found to reduce symptom severity in both ADHD and RLS [7,10,94,95].

### 4.4. Brain Iron Imaging

The measurement of brain iron via in vivo brain magnetic resonance techniques is the most direct approach to CID. Recent advances in MRI have opened up new opportunities, allowing safe and non-invasive brain iron level estimates in vivo [96]. Currently, two main methods are applied for measuring brain iron: (a) methods based on transverse relaxation times; and (b) methods based on phase information to produce susceptibility weighted imaging (SWI) and quantitative susceptibility maps (QSMs). Iron, especially when it is stored in ferritin or hemosiderin, induces local magnetic field inhomogeneities and increases relaxation rates. These rates can be measured by using a multi-echo acquisition, then performing voxelwise fitting to an exponential model [97]. SWI is a weighted MRI technique that enhances image contrast by using the susceptibility differences between tissues [98].

In pediatric patients, the early diagnosis of neurodegeneration with brain iron accumulation has been shown with SWI, and SWI has been integrated successfully into fetal MRI studies [99,100]. QSMs provide quantitative measures of magnetic susceptibility and are based on paramagnetic effects from iron-containing proteins that increase bulk tissue magnetic susceptibility. QSMs are calculated through a multi-step process that ultimately converts phase information acquired via a weighted sequence into magnetic susceptibility values [101]. QSMs have also been used to describe adult disorders that are associated with iron overload, such as neurodegenerative disorders and multiple sclerosis [102] and may support phenotyping studies for describing the role of iron homeostasis. While few imaging studies exist for ADHD, the results so far demonstrate lower brain iron levels, and the systematic review by Degremont et al. concluded that brain iron, rather than serum iron levels, tend to be lower in children with ADHD [103]. While most brain imaging studies in individuals with RLS have shown lower brain iron levels, a recent meta-analysis of 72 patients with RLS showed the normal and increased iron content of subcortical brain areas, showing the complexity of capturing affected iron levels in the brain [19]. One of the limitations of both transverse relaxation and phase-based techniques is the fact that they are not specific to iron. All matter has a diamagnetic or paramagnetic component to it and ultimately contributes to shifting the phase or transverse relaxation, with iron being one of the strongest contributors [104]. Thus, while an increase in paramagnetism would suggest an increase in iron, it could also be from a loss of diamagnetic tissue or an increase in some other paramagnetic component, for example. Furthermore, other aspects such as the iron oxidation state could further bias the quantification of iron with MRI [105]. Ultimately, the authors of the meta-analysis conclude that brain iron mobilization or homeostasis is impaired in RLS, possibly through a reduction in the functional availability of iron or as a function of a decreased prevalence of transferrin receptors [87,89]. These results raise questions, not only about the methodologies of imaging studies but also about how to capture pathophysiologic mechanisms that modulate iron homeostasis. Such contradictions have inspired the title of our research endeavor, the ‘Iron Conundrum’. To fully understand the role of iron in RLS, extensive post-mortem studies in large cohorts will be necessary, regardless of whether new methods to evaluate brain iron metabolism are developed in the future.

### 4.5. Possible Pathophysiological Mechanisms of Central Iron Deficiency

Animal studies utilizing dietary modifications provide strong evidence for the brain iron mobilization concept and demonstrate causal links between ID and alterations in dopamine transmission at multiple levels that are consistent with the hypothesis of a CID-induced dysregulation of dopamine receptors in RLS (Figure 1) [2]. The proposed mechanism is the impaired mobilization of iron in the brain, leading to a functional ID [19]. Connor et al. demonstrated, in a small sample of four patients with early-onset RLS, that transferrin receptor expression was decreased [106]. Further supporting this hypothesis of impacted iron homeostasis and CID are studies that have shown significantly increased hepcidin levels in individuals with ADHD and RLS compared to controls [107,108,109,110]. Hepcidin is the primary regulator of iron homeostasis and works to block the efflux of cellular iron. It is highly expressed during states of normal iron homeostasis and suppressed during states of systemic ID. Inflammatory states are associated with increased levels of hepcidin and, thus, decreased mobilization of iron to the plasma and absorption of iron from the gastrointestinal tract [1,110]. Altogether, these factors result in the insufficient incorporation of iron into erythroid precursors in the face of seemingly normal peripheral iron stores [60]. Importantly, animal studies have also implicated hepcidin as a key player of iron homeostasis in the brain [111]. Due to functional limitations on how to assess iron function in the spinal cord, the majority of clinical and animal studies have focused on ID-induced alterations in the brain, thereby giving rise to the term “brain iron deficiency” [2]. However, a more recent study indicates that the effects of low iron levels in a non-anemic state can also be detected by an increased expression of the transferrin receptor in the spinal cord and an augmented sensorimotor excitability [3], thus warranting the use of the wider term “central iron dysfunction”, rather than only referring to the brain.

### 4.6. Clinical Perspectives and Future Research

As stated in this review, most recommendations, including the WHO guidelines, have not incorporated conditions presenting with restlessness, such as ADHD and RLS, which opens many avenues for therapeutic interventions, a critical analysis of existing guidelines, and an evaluation of the suggested changes with future clinical and experimental research. The reality is that while the need to be aware of iron homeostasis and/or CID concepts has been acknowledged among clinicians and researchers working in the field of sleep medicine, these concepts have not yet been established in daytime-focused conditions such as ADHD, despite being a 24-hour disorder. Therefore, the interpretation of iron status in individuals with conditions such as ADHD and RLS still poses a unique challenge for clinicians.

The explorative neurophysiological characterization of ADHD sleep phenotypes and the clinical approach to RLS from a patient perspective have been important first steps of clinical phenotyping [112,113,114,115]. Now, we need a standardization of semiology to capture the same clinical phenotypes. Given that conditions presenting with restlessness are thought to be primarily due to impaired iron homeostasis and/or central iron dysfunction, peripheral iron levels (e.g., measured by SF) may not be helpful at guiding clinical decision making using existing ID cutoff values, particularly if we do not proactively investigate inflammation. The well-established RLS treatment guidelines that use SF cutoff values of 75 ug/L and 50 ug/L for adults (including pregnancy) and children, respectively [115,116,117], are in contrast to the majority of general ID guidelines that use much lower cutoff values for the diagnosis of absolute ID (Figure 5 and Figure 6). At this stage, we suggest a consensus approach for harmonized ID screening with a focus on inflammatory states and, second, the utilization of the RLS-specific SF cutoff values, as currently, ADHD-specific cutoff values do not exist. At present, universal screening for ID/IDA is not recommended in otherwise healthy children [118]. However, a recent study by Carsley et al. suggest that screening for ID in young children would be cost-effective, citing the effects of ID on development and cognitive function [119]. As sleep disturbances, particularly those presenting with restlessness (e.g., RLS, PLMS, RSD), often pose a challenge in clinical practice, we are in support of such a screening concept in children and adolescents with neurodevelopmental disorders and/or mental health conditions [7,10,120,121]. Whether this should be broadened to a universal screening concept in otherwise healthy children is a topic of further discussion. As mentioned above, cutoff values in the context of sleep disturbances and neurodevelopmental and/or mental health conditions have to be higher than for the diagnosis of absolute ID. The integration of ADHD and RLS into general ID guidelines will therefore catalyze this endeavor.

Finally, there appear to be several avenues for CID research through brain imaging. One such method involves decomposing the QSM signal into its diamagnetic and paramagnetic components. Iron and deoxygenated blood are highly paramagnetic and result in positive values in QSM, and diamagnetic tissue such as myelin result in negative values. When a voxel contains both diamagnetic and paramagnetic tissue, these values will subtract from each other, masking both kinds of tissue. Using a method such as DECOMPOSE-QSM or APART-QSM would allow for separate diamagnetic and paramagnetic maps, improving the sensitivity and specificity for iron [122,123]. Another avenue is the fact that, previously, most clinically acquired SWI scans did not save the unfiltered phase images. QSM requires an unfiltered phase image in order to be calculated. Recent studies, using deep learning algorithms, have been trained to recover the unfiltered phase information from the filtered phase data [19,124]. Thus, many previous studies that could not be used to calculate QSM can now be carried out retrospectively, resulting in a potentially vast amount of ID data to be analyzed. In summary, the inclusion of QSM in MRI studies of patients with restlessness and a review of iron status might be the next avenue to explore in future studies.

## 5. Limitations

Despite conducting a scoping literature review, most guidelines were identified by searching the gray literature, which consisted of Google searches, searching websites of international, national, and regional health organizations, and utilizing the Trip Medical Database to identify guidelines that met our inclusion criteria. Thus, it is very likely that additional guidelines exist and were not included in this review, which could result in changes to our discussion and conclusions. Similarly, guidelines not published in English were excluded. Due to the fact that many of the included guidelines were found on websites, there is also a possibility that guidelines may be removed and/or updated, with the original guideline no longer available. In addition, our results show that the methodology for guideline development varied, with over half of the included guidelines not providing an explanation of their methods; this lack of transparency may have implications for guideline rigor and accuracy. However, this topic is outside the scope of this review and, therefore, was not analyzed in more detail.

One further limitation is the lack of accurate semiology. After in-depth discussions, the authors decided to use the term ‘restlessness’ as an umbrella term. However, as restlessness alters tone and posture, both of which are indicative of vigilance alterations, we may need to revisit clinical phenotyping terminology using the descriptions of visible motor and vigilance states for sleep and wake behaviors. This would then allow for capturing a broader spectrum of movement patterns; on the one side, hyperkinesia, presenting with a hypermotor and hypo- or hypervigilant state [112,125], and on the other hand, with hypokinesia, presenting with hypo- or hypervigilance [126], but also neurodiverse patients with restrictive hypokinesia/hypomotor behaviors [127]. Given the lack of consensus on phenotyping terminology, our provocative simplification might also be considered as a strength, as it starts the necessary semiology discourse.

## 6. Conclusions

The existing ID guidelines analyzed in this scoping review highlight the gaps in clinical knowledge, particularly when it comes to the concept of CID and restlessness. The majority of guidelines do not include ADHD or RLS. Further, the heterogeneity of biomarkers and SF cutoff values raises questions about the application of current ID guidelines in clinical practice. Given this, the next steps will be to (a) achieve consensus on how to investigate and interpret iron status as part of a homeostatic system (affected by age, sex, pregnancy, inflammation, and comorbidities), (b) develop screening criteria for conducting iron studies, (c) incorporate the concept of iron homeostasis and CID into clinical practice guidelines, and (d) conduct further brain imaging studies to better elucidate the role of iron in the pathophysiology of conditions presenting with restlessness in sleep and wake states.

## Figures and Tables

**Figure 1 nutrients-16-02559-f001:**
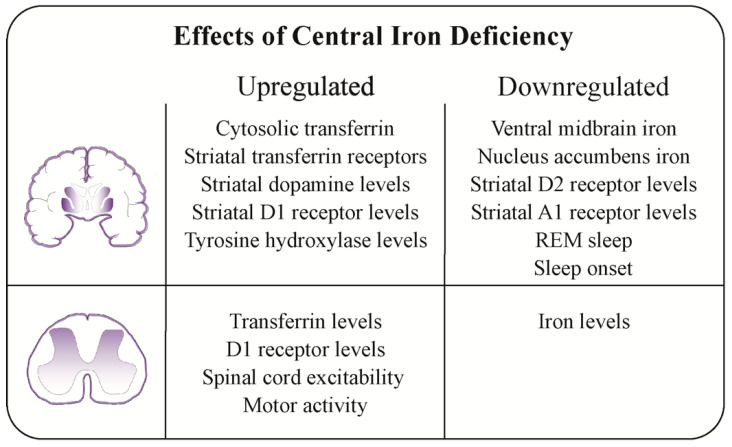
Effects of central iron deficiency (CID), as grouped by region of interest, brain, and spinal cord. CID refers to low iron levels in both the brain and spinal cord. Highlighted here are areas in the brain and spinal cord that are particularly susceptible to CID-induced changes in dopamine modulation. In the striatum, they are suspected to modulate motor outputs, while in the dorsal spinal cord, they are associated with sensory inputs. Data compiled from Silvani et al. [2] and Woods et al. [3]. A1 receptor: adenosine receptor; D1/D2 receptor: dopamine receptors; REM: rapid eye movement.

**Figure 2 nutrients-16-02559-f002:**
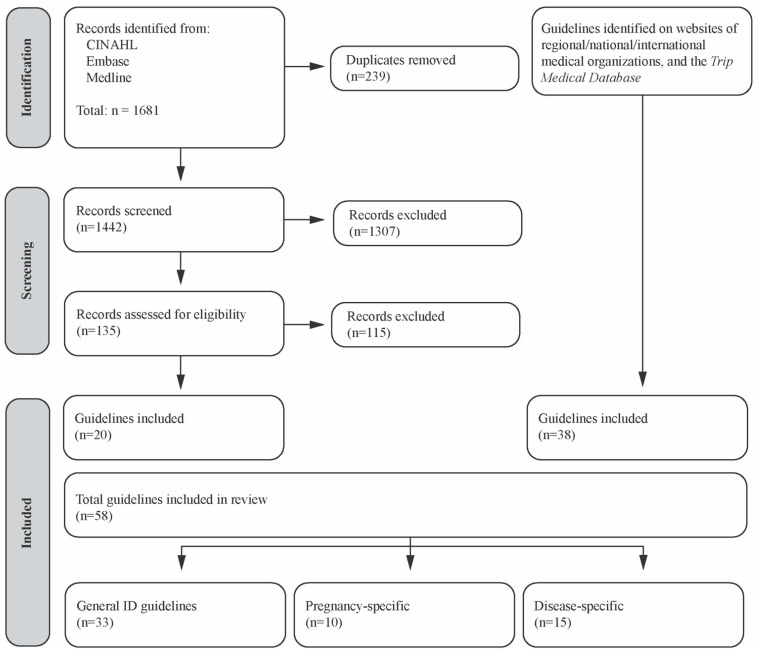
Study selection. Adapted from [88].

**Figure 3 nutrients-16-02559-f003:**
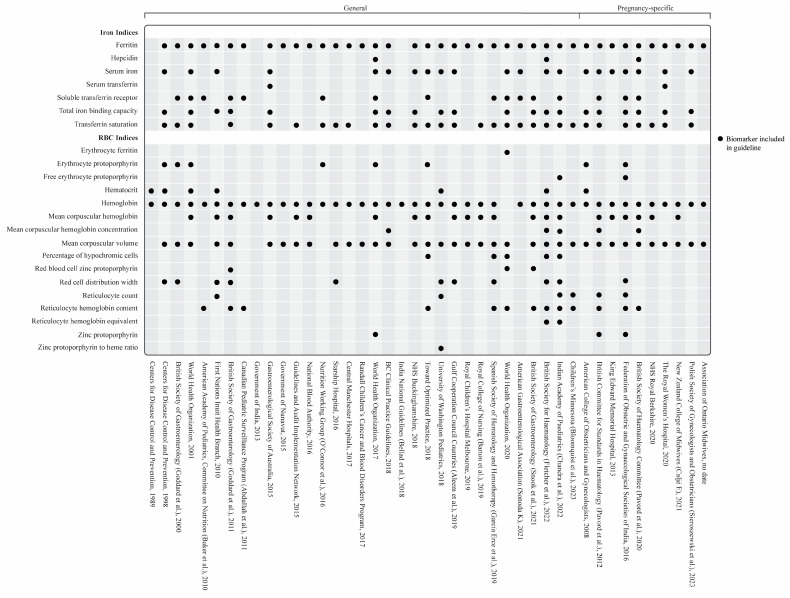
Recommended biomarkers for the diagnosis of NAID/IDA in the included general and pregnancy-specific guidelines [30,31,32,33,34,35,36,37,38,39,40,41,42,43,44,45,46,48,51,52,53,54,55,57,58,64,65,67,68,70,71,72,73,74,75,76,77,78,80,81,85,86,87].

**Figure 4 nutrients-16-02559-f004:**
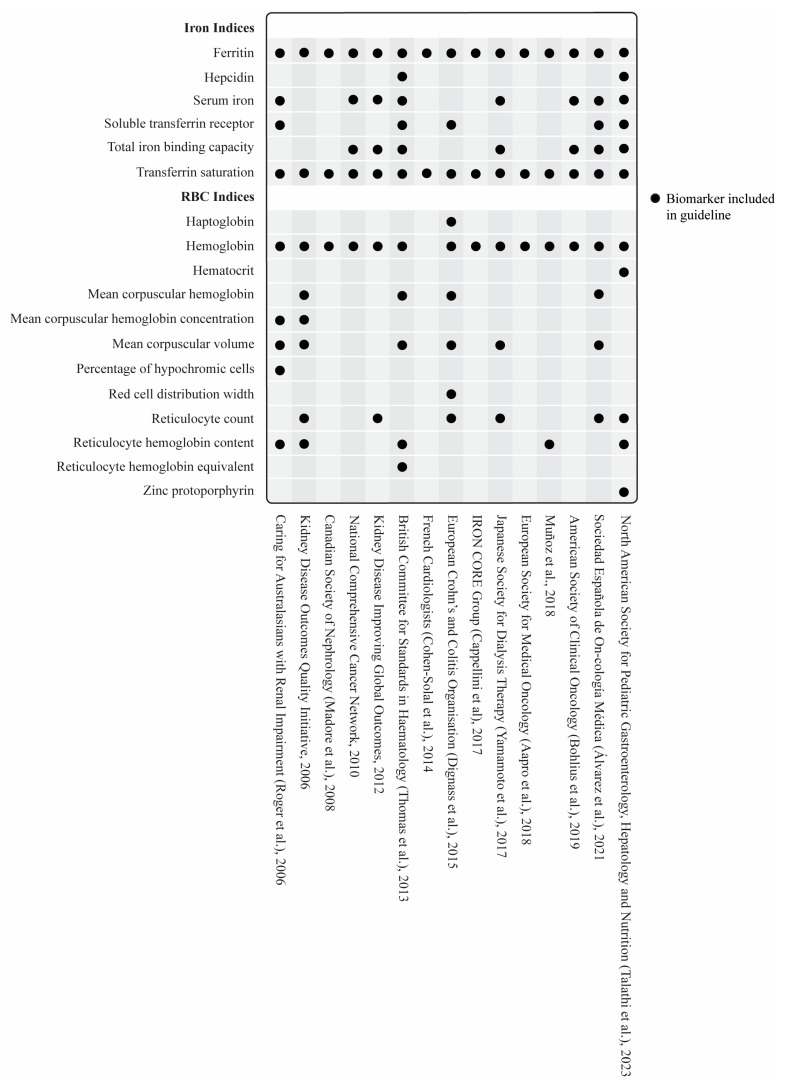
Recommended biomarkers for the diagnosis of NAID/IDA in the included disease-specific guidelines [47,49,50,56,59,60,61,62,63,66,69,79,82,83,84].

**Figure 5 nutrients-16-02559-f005:**
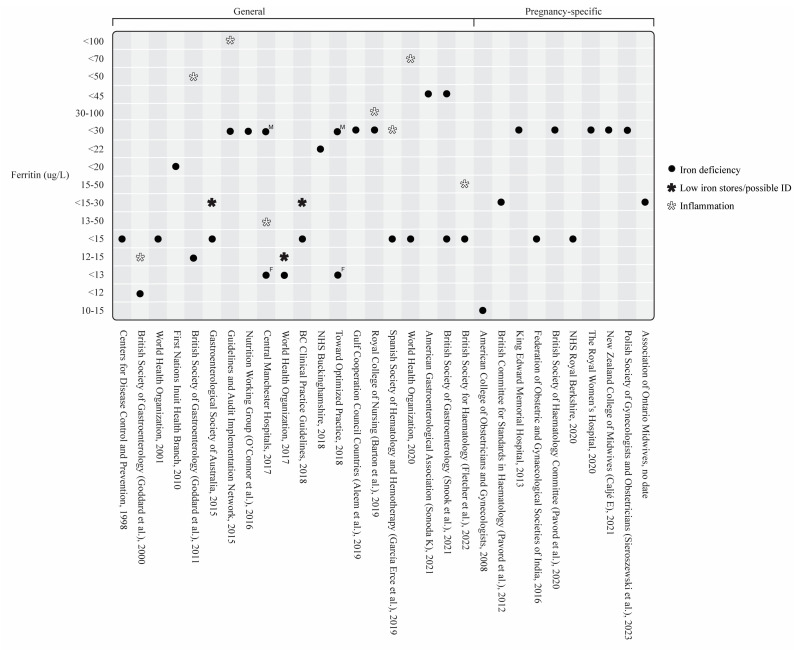
Serum ferritin cutoff values of the included general and pregnancy-specific guidelines. F: female; M: male [30,31,33,34,35,39,40,42,43,44,45,46,51,53,54,55,57,67,68,70,71,72,74,75,77,78,80,81,85,87].

**Figure 6 nutrients-16-02559-f006:**
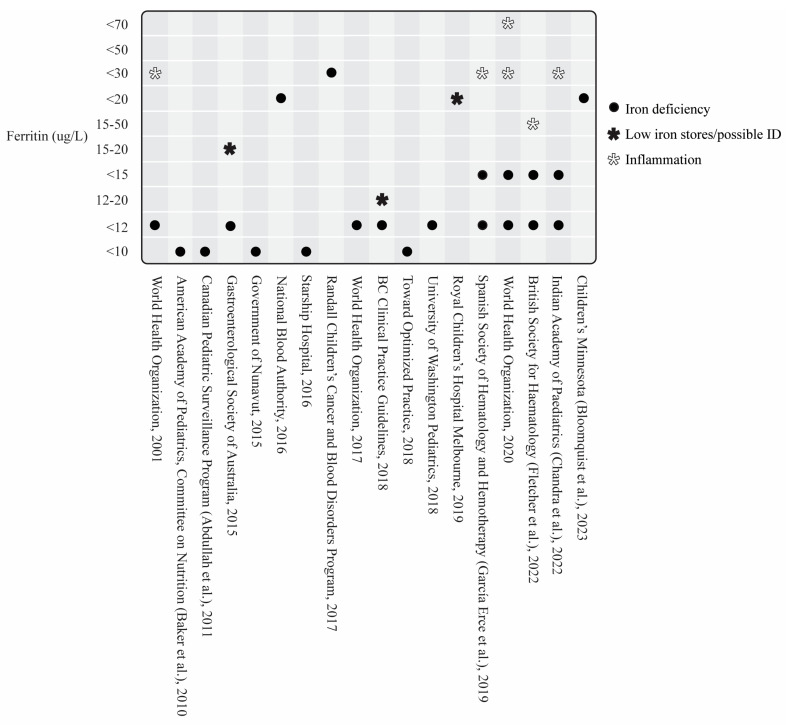
Pediatric-specific serum ferritin cutoff values of the included general ID guidelines [30,31,35,37,38,40,41,42,43,48,52,58,64,65,72,73,74,76].

**Table 2 nutrients-16-02559-t002:** Search strategy used in CINAHL, Embase, and Medline.

Database		
CINAHL	(MH “Anemia, Iron Deficiency”) OR (TI “iron deficiency anemia” OR AB “iron deficiency anemia”) OR (TI “iron deficiency” OR AB “iron deficiency”)	(MH “Practice Guidelines”) OR (TI guideline or AB guideline)
Embase	(iron deficiency/) OR (iron deficiency anemia/) OR (iron deficiency anemia. tw, kw.) OR (iron deficiency. tw, kw.)	(practice guideline/) OR (guideline. tw, kw.)
Medline	(Anemia, Iron Deficiency/) OR (iron deficiency. tw, kf.) OR (iron deficiency anemia. tw, kf.)	(Practice Guideline/OR Guideline/) OR (Guideline. tw, kf.)

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
