# Peer review of "Iron Deficiency and Sleep/Wake Behaviors: A Scoping Review of Clinical Practice Guidelines—How to Overcome the Current Conundrum?"

_nutrients, 2024, doi:10.3390/nu16152559_

Round 1
Reviewer 1 Report
Comments and Suggestions for Authors
The authors have explored a topic of considerable importance and impact in the clinical practice of these pathologies in adults and children, also considering important aspects in basic research. The review is complete and well structured, the tables exhaustive and very explanatory. I congratulate the authors and have no further suggestions.
Author Response
Comments: The authors have explored a topic of considerable importance and impact in the clinical practice of these pathologies in adults and children, also considering important aspects in basic research. The review is complete and well structured, the tables exhaustive and very explanatory. I congratulate the authors and have no further suggestions.
Response: Thank you for your appreciated comments.
Reviewer 2 Report
Comments and Suggestions for Authors
Thank you very much on your review.
I have few, very minor comments that would I hope lead to better readability.
Figure 1 needs better explanation under, so it is able to exist as standalone figure. I had to refer back to text to understand it.
Sentence "While specific mechanisms by which NAID contributes..." that mentions Table 1 for the first time in Introduction needs a bit reflection on size of the study samples and inconsistency of results (inconsistency bit can be tied in a bit better with the last sentence in Introduction).
Table 3, 4 and 5 are perhaps better suited for appendix.
I also have to comment on Figures 3-6; they are so good, I haven't seen anything like that before.
Best regards
Author Response
Thank you very much for taking the time to review our manuscript. All changes have been highlighted within the manuscript document. Below is a summary of our responses.
Comment #1: Figure 1 needs better explanation under, so it is able to exist as standalone figure. I had to refer back to text to understand it.
Response: We have expanded the figure caption to be more descriptive:
Figure 1. Effects of central iron deficiency (CID), as grouped by region of interest, brain and spinal cord. CID refers to low iron levels in both the brain and spinal cord. Highlighted here are areas in the brain and spinal cord that are particularly susceptible to CID-induced changes in dopamine modulation. In the striatum, they are suspected to modulate motor outputs, while in the dorsal spinal cord, they are associated with sensory inputs. Data compiled from: Silvani et al. [2] and Woods et al. [3]. A1 receptor: adenosine receptor; D1/D2 receptor: dopamine receptors; REM: rapid eye movement.
Comment #2: Sentence "While specific mechanisms by which NAID contributes..." that mentions Table 1 for the first time in Introduction needs a bit reflection on size of the study samples and inconsistency of results (inconsistency bit can be tied in a bit better with the last sentence in Introduction).
Response: To address reviewer #3’s comment, along with this one, we have incorporated a brief explanation within the discussion (page 15) on some of the limitations of the imaging techniques utilized in the studies represented in Table 1. We have not made an additional paragraph about the sample sizes, in order not to lengthen the discussion any further. If the reviewer thinks this information is absolutely necessary, we are most happy to add that.
One of the limitations of both transverse relaxation and phase-based techniques is the fact that they are not specific to iron. All matter has a diamagnetic or paramagnetic component to it and ultimately contributes to shifting the phase or transverse relaxation, with iron being one of the strongest contributors [103]. Thus, while an increase in paramagnetism would suggest an increase in iron, it could also be from a loss of diamagnetic tissue, or the increase of some other paramagnetic component, for example. Furthermore, other aspects such as the iron oxidation state could further bias the quantification of iron with MRI [104]. Ultimately, the authors of the meta-analysis conclude that brain iron mobilization or homeostasis is impaired in RLS, possibly through a reduction in the functional availability of iron or as a function of a decreased prevalence of transferrin receptors [87,88]. These results raise questions, not only about methodologies of imaging studies but also about how to capture pathophysiologic mechanisms that modulate iron homeostasis. Such contradictions have inspired the title of our research endeavor, the ‘Iron Conundrum’. To fully understand the role of iron in RLS, extensive post-mortem studies in large cohorts will be necessary, regardless of whether new methods to evaluate brain iron metabolism are developed in the future.
Comment #3: Table 3, 4 and 5 are perhaps better suited for appendix.
Response: Tables 3-5 have been moved to the appendix (now Tables A2-A4; page 20-32).
Comment #4: I also have to comment on Figures 3-6; they are so good, I haven't seen anything like that before.
Response: Thank you for this encouraging comment; it took us some time to come up with the concept, so we are delighted to hear this nice feedback. Thank you again!
Reviewer 3 Report
Comments and Suggestions for Authors
Dear Corresponding Author,
thank you for submitting your article and congratulations on your work.
The article provides a useful systematic review on the effects of a specific dietary intervention on athletic performance. The structure follows the PRISMA-ScR guidelines, ensuring a methodologically rigorous analysis. The review clearly identifies trends and gaps in the existing literature, providing useful indications for future research.
I anticipate that the work is definitely interesting and deserves publication, however I suggest some changes after careful reflection which I extend below following the guidelines of Nutrients journal:
1. Brief summary: This article presents a comprehensive and well-structured scoping review of existing clinical guidelines on iron deficiency (ID). The main objective of examining diagnostic recommendations, iron biomarkers, cut-off values and the role of inflammation in interpreting laboratory results has been successfully achieved. Additionally, the identification of the inclusion of common disorders such as ADHD and RLS in clinical guidelines provides an important perspective on the clinical management of ID.
2. General comments: The work offers a significant contribution to the existing literature, highlighting important gaps in current ID guidelines. The methodology used is appropriate for a scoping review, following the PRISMA-ScR guidelines. However, more details could be provided on the search strategy and study selection process. The data analysis is complete and well presented, with informative tables and figures that effectively support the text. The discussion is well structured and offer a critical interpretation of the results, linking them to existing literature and implications for clinical practice. The article could benefit from a more detailed description of the inclusion and exclusion criteria used to select studies. In addition, a more in-depth analysis of the aggregated results would be useful to provide a clearer understanding of the impact of the dietary intervention. Finally, since some references are linked to very old publication, they could be updated to include more recent studies and improve impact. Also review that all references have the DOI.
3. Specific comments:
· Line 45: Clarify why some studies were excluded from the review
· Line 101-102 In the Introduction:. After "...in clinical practice.", a brief explanation could be added as to why a scoping review was chosen instead of a traditional systematic review.
· Line 115-116. In Methods: After "Two reviewers (SM and OH) carried out data extraction.", section 2.1 "Data Analysis" could be expanded to provide more details on how the analysis of data extracted from the guidelines was conducted.
· Line 146-149. In Results: After "Three guidelines did not specify which grading system was used [63,66,68].", a brief discussion of the implications of methodological differences between guidelines that provided a classification of evidence and those that did not would be useful.
· Line 391-392 In Discussion:. After "...showing the complexity of capturing affected iron levels in the brain [87].", you might consider adding a brief discussion of the potential limitations of these imaging techniques and how they might influence the interpretation of the results of the studies cited.
· Line 494-496. In Conclusions: After "...with future clinical and experimental research.", the conclusions could be strengthened by providing more specific recommendations for future research, in addition to suggesting a "new consensus".
· Figure 1: Add a brief legend below the figure to explain the abbreviations used, to improve understanding for readers less familiar with the specific terminology.
· Figure 3: Provide a more detailed legend to facilitate data interpretation.
· Table 2: Add details on the characteristics of participants in the included studies.
· Tables 3-5: At the end of each table, consider adding a footnote to explain the less common abbreviations used.
Overall, this is a high-quality work that provides an important overview of the current state of ID guidelines and identifies key areas for improvement and future research. With some minor revisions, it will be a valuable contribution to the literature and i look forward to rereading it with the suggested changes or those you will have considered applicable after your critical reflection.
Author Response
Our apologies, however, we believe that, possibly due to high workload, some of the comments for the review of a manuscript about dietary intervention on athletic performance were mixed up with the review of our manuscript.
We assume that the following comments were not intended for our manuscript, though we have attempted to address them where possible:
Note: all changes have been highlighted within the manuscript document.
Comment: some references are linked to very old publication
Response: Some of the references that are linked to older publications (e.g., from the 1980’s and 90’s) are guidelines that are included in this review, and therefore must stay. In addition, there are some older studies from the 2000’s on brain iron imaging that we have referenced; given the overall limited number of brain iron imaging studies, we feel it appropriate and necessary to incorporate as much of the existing literature as possible, including those older studies.
Comment: a more in-depth analysis of the aggregated results would be useful to provide a clearer understanding of the impact of the dietary intervention
Response: Our review did not look at the impact of dietary intervention, thus we believe this comment was not intended for our manuscript.
Comment: Line 45: Clarify why some studies were excluded from the review.
Response: Line 45 seems to correspond to the abstract, therefore it was not clear exactly what this comment was referring to. Nonetheless, we have expanded on the methods to now include a more detailed section on inclusion and exclusion criteria on page 6.
Comment: Figure 3: Provide a more detailed legend to facilitate data interpretation.
Response: We are not sure if this is related to our manuscript; nonetheless we have added a brief legend to Figures 3 and 4 to explain what the black dots mean. Thank you.
Comment: Table 2: Add details on the characteristics of participants in the included studies.
Response: Table 2 is referring to the search strategy.
Comment: Tables 3-5: At the end of each table, consider adding a footnote to explain the less common abbreviations used.
Response: We believe this comment may be referring to a different review as we have included all abbreviations in the table caption (note tables 3-5 have now been moved to the appendix based on another reviewer's comment; they are now tables A2-A4).
Comment: Line 494-496. In Conclusions: After "...with future clinical and experimental research.", the conclusions could be strengthened by providing more specific recommendations for future research, in addition to suggesting a "new consensus".
Response: The sentence referred to in this comment was in the discussion, not the conclusion of our manuscript. We believe that our conclusions are quite concrete. If the reviewer still thinks that more specific recommendations are needed, we would be happy to adjust our conclusions.
Additional comments:
Comment: Also review that all references have the DOI
Response: All references have been updated to include a DOI where applicable.
Comment: Line 101-102 In the Introduction:. After "...in clinical practice.", a brief explanation could be added as to why a scoping review was chosen instead of a traditional systematic review.
Response: Thank you! We have incorporated a brief explanation at the end of the introduction on page 5.
A scoping review was the agreed upon methodology given the exploratory nature of this project and to facilitate the identification of gaps in clinical knowledge.
Comment: Line 115-116. In Methods: After "Two reviewers (SM and OH) carried out data extraction.", section 2.1 "Data Analysis" could be expanded to provide more details on how the analysis of data extracted from the guidelines was conducted.
and
Comment: The article could benefit from a more detailed description of the inclusion and exclusion criteria used to select studies
Response: Thank you! We have expanded on the methods, including section 2.1 (which is now section 2.3 after creating two additional sections for inclusion and exclusion criteria).
- Materials and Methods
This scoping review was carried out in accordance with the PRISMA-Scr Checklist (Appendix A, Table A1). A protocol for this review does not exist. Two reviewers (SM and OH) were involved in identifying guidelines for inclusion. The initial search was carried out on June 27, 2020 in CINAHL, Embase, and Medline with no date restrictions, and was updated on April 7, 2023. The search strategy included variations of the terms “iron deficiency” and “guideline”; the detailed search strategy is laid out in Table 2. Covidence was employed for the selection and de-duplication process. The search was updated in 2023. Additional guidelines were identified by searching Trip Medical Database (https://www.tripdatabase.com/) as well as conducting Google searches and checking websites of medical organizations. Two reviewers (SM and OH) carried out data extraction and organized data into a spreadsheet. A third reviewer (OI) was available to oversee this process.
2.1. Inclusion Criteria
Guidelines were included if they were (1) general ID guidelines, defined as those which targeted a general population (e.g., adults, elderly, children, ethnic groups), pregnancy-specific ID guidelines, and disease specific guidelines (e.g., chronic kidney disease (CKD)); (2) if the guideline or consensus paper was created by/on behalf of a larger governing body (e.g., international, national or regional organizations/societies); (3) available in English-language.
2.2. Exclusion Criteria
Guidelines were excluded if they were (1) opinion papers or guidelines published by authors not affiliated with a larger governing body; (2) reviews of clinical guidelines.
2.3. Data Analysis
Three reviewers (SM, OH, and OI) were involved in analyzing the extracted data. A qualitative analysis was carried out by reviewing the following categories:
- Population defined by age, geographical region, pregnancy, medical conditions. Guidelines were organized into three categories: 1) general ID, 2) ID in pregnancy, and 3) disease-specific ID.
General ID guidelines were defined as those guidelines which could be applied to a general population, and which may have included specific subpopulations within the guideline.
Disease-specific ID guidelines were defined as those guidelines which dealt with only with a specific population, namely chronic disease populations, in which the diagnosis and management of ID is different from general ID guidelines.
- Year and country of publication.
- Associated clinical presentations, conditions, diagnoses, and risk factors for ID.
- If ADHD and/or RLS were included as either signs/symptoms, or as being associated with ID. Guidelines that used broad terminology such as “behavioural disturbances” or “sleep disturbances” without specifying the aforementioned conditions, were not classified as having included either ADHD and/or RLS
- Suggested cutoff values for SF, taking into account age and sex-specific cutoff values.
- 3.1 Additional iron and hematologic biomarkers included in the guidelines. Examples of iron biomarkers (other than SF) are serum iron and transferrin saturation, while hematologic biomarkers include hemoglobin and mean corpuscular volume/mean corpuscular hemoglobin.
Comment: Line 146-149. In Results: After "Three guidelines did not specify which grading system was used [63,66,68].", a brief discussion of the implications of methodological differences between guidelines that provided a classification of evidence and those that did not would be useful.
Response: Thank you. We have included this within the limitations section on page 17.
In addition, our results show that the methodology for guideline development varied, with over half of the included guidelines not providing an explanation of their methods; this lack of transparency may have implications for guideline rigor and accuracy, however this topic is outside the scope of this review and therefore was not analyzed in more detail.
Comment: Line 391-392 In Discussion:. After "...showing the complexity of capturing affected iron levels in the brain [87].", you might consider adding a brief discussion of the potential limitations of these imaging techniques and how they might influence the interpretation of the results of the studies cited.
Response: We have incorporated some of the limitations of the brain imaging techniques on page 15.
One of the limitations of both transverse relaxation and phase-based techniques is the fact that they are not specific to iron. All matter has a diamagnetic or paramagnetic component to it and ultimately contributes to shifting the phase or transverse relaxation, with iron being one of the strongest contributors [103]. Thus, while an increase in paramagnetism would suggest an increase in iron, it could also be from a loss of diamagnetic tissue, or the increase of some other paramagnetic component, for example. Furthermore, other aspects such as the iron oxidation state could further bias the quantification of iron with MRI [104]. Ultimately, the authors of the meta-analysis conclude that brain iron mobilization or homeostasis is impaired in RLS, possibly through a reduction in the functional availability of iron or as a function of a decreased prevalence of transferrin receptors [87,88]. These results raise questions, not only about methodologies of imaging studies but also about how to capture pathophysiologic mechanisms that modulate iron homeostasis. Such contradictions have inspired the title of our research endeavor, the ‘Iron Conundrum’. To fully understand the role of iron in RLS, extensive post-mortem studies in large cohorts will be necessary, regardless of whether new methods to evaluate brain iron metabolism are developed in the future.
Comment: Figure 1: Add a brief legend below the figure to explain the abbreviations used, to improve understanding for readers less familiar with the specific terminology.
Response: Thank you! We have included the abbreviations within the figure caption.
Figure 1. Effects of central iron deficiency (CID), as grouped by region of interest, brain and spinal cord. CID refers to low iron levels in both the brain and spinal cord. Highlighted here are areas in the brain and spinal cord that are particularly susceptible to CID-induced changes in dopamine modulation. In the striatum, they are suspected to modulate motor outputs, while in the dorsal spinal cord, they are associated with sensory inputs. Data compiled from: Silvani et al. [2] and Woods et al. [3]. A1 receptor: adenosine receptor; D1/D2 receptor: dopamine receptors; REM: rapid eye movement.